# The Use of Stable Isotope Ratio Analysis to Trace European Sea Bass (*D. labrax*) Originating from Different Farming Systems

**DOI:** 10.3390/ani10112042

**Published:** 2020-11-05

**Authors:** Francesca Tulli, José M. Moreno-Rojas, Concetta Maria Messina, Angela Trocino, Gerolamo Xiccato, José M. Muñoz-Redondo, Andrea Santulli, Emilio Tibaldi

**Affiliations:** 1Department of Agriculture, Food, Environment and Animal Sciences, University of Udine, Via Sondrio 2, 33100 Udine, Italy; francesca.tulli@uniud.it (F.T.); emilio.tibaldi@uniud.it (E.T.); 2Department of Food Science and Health, Andalusian Institute of Agricultural and Fisheries Research and Training (IFAPA), Alameda del Obispo Centre, Avda. Menendez Pidal, s/n, 14004 Córdoba, Spain; josem.munoz.redondo@juntadeandalucia.es; 3Laboratory of Marine Biochemistry and Ecotoxicology, Department of Earth and Sea Science, University of Palermo, Via Barlotta, 4, 91100 Trapani, Italy; concetta.messina@unipa.it (C.M.M.); andrea.santulli@unipa.it (A.S.); 4Department of Comparative Biomedicine and Food Science, University of Padova, Viale dell’Università 16, 35020 Legnaro, Padova, Italy; angela.trocino@unipd.it; 5Department of Agronomy, Food, Natural Resources, Animals and Environment, University of Padova, Viale dell’Università 16, 35020 Legnaro, Padova, Italy; gerolamo.xiccato@unipd.it

**Keywords:** aquaculture, *Dicentrarchus labrax*, stable isotopes, traceability, farming system, geographic origin, IRMS, sea bass, fish, authentication

## Abstract

**Simple Summary:**

European sea bass is one of the most economically important fish species in the Mediterranean area. The potential effects of farming systems on the final quality of this product and the recent popular demand for labels to certify the animal rearing origin, which is increasingly used as a marketing tool, have raised the use of analytical techniques that make it possible to differentiate this fish product according to the rearing farming system and authenticate their geographical origin. The aim of this study was to determine whether isotopic ratio mass spectrometry (IRMS) can discriminate farmed European sea bass according to different farming systems (concrete tank inland, sea cages, and extensive methods in valleys or salt works) and geographic origins (different locations scattered throughout Italy). The results of this study showed the viability of δ^13^C and δ^15^N to discriminate cultured sea bass from different farming systems (extensive vs. intensive) reared at different geographical sites in Italy. Meanwhile, the measurement of δ^18^O and δ^2^H made it possible to distinguish the geographical origin of the sea bass farmed extensively and intensively (in cages).

**Abstract:**

This study aimed to determine whether isotopic ratio mass spectrometry (IRMS) can discriminate farmed European sea bass according to different farming systems and geographic origins. *Dicentrarchus labrax* of commercial size from three different rearing systems (concrete tank inland, sea cages, and extensive methods in valleys or salt works) were collected at the trading period (autumn–winter). For each farming type, different locations spread over Italy were monitored. Once the fish were harvested, the muscle and feed were sampled. For both muscle and feed, δ^13^C and δ^15^N were measured by continuous flow elemental analyzer isotope ratio mass spectrometry (CF-EA-IRMS) with the goal of discriminating samples based on the rearing system. Additional δ^2^H and δ^18^O measurements of fish samples were performed by continuous flow total combustion elemental analyzer isotope ratio mass spectrometry (CF-TC/EA-IRMS) to track the geographical origin. The measurements of δ^13^C and δ^15^N made it possible to discriminate cultured sea bass from different farming systems (extensive vs. intensive) reared at different geographical sites in Italy. Additional information was obtained from δ^18^O and δ^2^H, which enabled the geographical areas of origin of the sea bass farmed extensively and intensively (in cages) to be distinguished.

## 1. Introduction

In recent years, the trade of seafood has experienced strong growth due to the rising world population and increases in the annual consumption per capita of seafood from 9.9 kg in the 1960s to over 20 kg in 2013 [1]. In 2013, fish accounted for 17% of the global population’s intake of animal protein and 6.7% of all protein consumed [1]. Consequently, international trade of fish products has addressed the evolution of food safety and food quality issues as emphasized by several EU Directives introduced into the chain for fisheries and aquaculture products with the concept “from farm to fork” usually based on the Codex Alimentarius provisions. In addition, national and transnational regulations regarding meat traceability have been imposed (European Regulation (EU) n. 1379/2013) to ensure more accessible details for retailers and consumers about labeling, packaging, and origin of wild capture and aquaculture products. In this respect, consumers are more and more concerned about the origin of foodstuffs for both health and ecological reasons, and food quality has become an essential parameter for their consumer preferences. Therefore, product differentiation appears to be a fundamental issue for the further development of the fish farming industry distributed around the whole Mediterranean area involved in different rearing and environmental systems.

The potential effects of different fish farming systems involve modifications to the final product quality such as fish appearance, organoleptic properties or even nutritional characteristics [2]. Feeding and breeding conditions are known to affect the nutritional quality of the fillet, the high content in essential polyunsaturated omega 3 and omega 6 fatty acids of wild fish and suitably fed farmed fish being of note [3]. Linked to, but different from, consumers’ demand for fish quality standards is the public perception that aquaculture can harm the environment [1]. In recent years, intensive farming has led to misperceptions and mistrust among consumers. Thus, authorities have acknowledged the need for labels that certify animal health and welfare, food safety and quality, environmental integrity, and social responsibility associated with aquaculture. The aims of such labels are (i) to reassure producers, buyers, consumers and civil society regarding the quality and safety of aquaculture products, and (ii) to provide a further tool to support responsible and sustainable aquaculture. A popular demand for labels to certify the animal rearing origin has emerged over the last years since, and, according to formal regulation, it is increasingly used as a marketing tool rather than a designation of quality and safety. Thus, besides being a tool to guarantee food safety, tracking, and traceability, these labels are also of major interest to retail business as a powerful communication tool aimed at improving consumer confidence [4,5,6,7,8,9,10]. However, food products with geographical indications and designation of origins following the European Regulation (EU) n. 1151/2012 are generally expensive but bring greater benefits to the producers than ordinary products. In this sense, consumers seem to be prepared to accept higher prices linked to superior quality, certification of the production process and the product origin.

European sea bass (*Dicentrarchus labrax* L.) is one of the most economically important fish species in the Mediterranean area [11] as it is one of the most requested marine species for its overall quality, with an interesting polyunsaturated fatty acid (PUFA) content 0.58 g/100 g of docosahexaenoic acid (DHA, 22:6(*n* − 3)) and 0.44 g/100 g of eicosapentaenoic acid (EPA, 20:5(*n* − 3)) (EC, 2015 [12]). Sea bass is found in the Mediterranean area in a variety of different culture systems including highly intensive recirculating systems, flow through concrete raceways or pond systems, floating sea cages, as well as traditional extensive pond systems located in highly environmentally sensitive areas [2]. Currently, intensive systems are the most common rearing method in this area due to a higher production yield [13]. Many publications have considered the authentication and quality of sea bass in relation to different factors, such as eco-physiological factors, diet, rearing condition, and differences between wild and farmed fish, but the reported results refer to a limited period of time, a restricted geographical area, or to only one type of culture system compared to the wild [2,14,15,16,17,18,19,20,21,22,23].

Despite the attention given to these issues, mislabeling, whether accidental or fraudulent, is expected to occur [24]. In the last few years, an increase in such mislabeling concerning the product processing and fish origin has been reported [25]. Therefore, new tools enabling simple and accurate discrimination between farming systems and authenticating their geographical provenance would be extremely valuable. Stable isotope ratio mass spectrometry (S-IRMS) could be used as an alternative tool to PCR-DGGE techniques that have been recently proposed to indirectly discriminate the geographical origin of fish by the analysis of the DNA fragments of microorganisms [26]. The isotopic content of an animal’s diet is known to affect the isotopic ratio of its meat [27,28,29]. For this reason, IRMS has previously been used for ecosystem studies [30,31], for fish and shellfish specifically, food web assessment [32,33,34,35], and back calculations of diet [36,37]. The technique was also used to enable geographical sourcing for plants and animals [38,39,40,41]. The ability of IRMS to thoroughly characterize a sample and therefore accurately discriminate different samples, makes it a powerful forensic tool to detect fraud [42,43].

In general, animals, and, in particular, fish are a complex substrate for the interpretation of isotopic enrichment as the abundance of stable nitrogen (δ^15^N) and stable carbon (δ^13^C) isotopes are the result of both the feed ingested as well as the fractionation occurring through the metabolic processes. The first application of stable isotope analysis in fish product authentication is quite recent [44]. Based on the assumption of the different isotopic signature of the feeding relationships in aquatic environments between natural food webs and fish reared on feed, wild and farmed Atlantic salmon were successfully discriminated by the relative abundances of N and C isotopes in their fillets. This technique has been successively proposed for the unequivocal discrimination of wild and cultured sea bass [14,45], sea bream [46,47], shrimp [48], other fish [49,50], or even for the differentiation of different species in the same family (gadoids) [51]. In contrast, only a limited number of applications are available for differentiating cultured fish farmed according to different types of diet [37,52,53], farming system or different regions of a relatively small area [54]. δ^13^C and δ^15^N have demonstrated potential for the geographical discrimination of wild fish [55], but limitations exist as it depends on the feeding habits of the animals. In this regard, the ^2^H/^1^H and ^18^O/^16^O ratios have been proven to be very useful to trace the geographical origin of a food product as their abundance largely reflects climatic differences depending on temperature, latitude, altitude, and distance from the sea [56,57]. According to this, the measurement of the stable isotope ratios of hydrogen and oxygen are applicable to the characterization of geographical origin because they are strongly latitude dependent. This was the case of different food products from different countries previously reviewed by [58]. Therefore, isotope analysis is considered to be an excellent tool for origin assessment [59] in the specific case of fish products due to the global nature of production and the variable point of origin, and also taking into account the different rearing systems in use, this tool appears to be of even more interest.

This study aimed to determine whether isotopic ratio mass spectrometry (IRMS) makes it possible to discriminate farmed European sea bass raised in extensive or intensive systems. For that purpose, the muscle of three types of commercially farmed sea bass from different systems (intensively inland basin, intensively sea cages in the sea, and extensive coastal lagoon) and locations in Italy were sampled to determine whether it was possible to discriminate the rearing systems and geographical origins.

## 2. Materials and Methods

### 2.1. Description of Experimental Design, Fish Sampling and Farms

We purchased the fishes uses in this study from the commercial farms. The farms used the procedures according to the commercial standards and regulations (Reg. (CE) N. 1099/2009).

One-hundred-seventy-five European sea bass (*Dicentrarchus labrax*) were sampled for one month (December–January) from 11 aquaculture plans (10–20 specimens per farm) representative of different farming systems: extensive (E; *n* = 4), intensive in sea cages (C; *n* = 3), and intensive inland (I; *n* = 4) farms. The location and the main characteristics of the selected farms are reported in Table 1 (map, Appendix A). One of the extensive farms (E3) also used commercial feed, as supplier declared after additional requests on our part once we obtained the isotopic data. Commercial fish from the suppliers were slaughtered in an ice slurry according to the commercial standards of each farm, put in polystyrene boxes covered with ice, and stored at the university laboratories at 24 °C until processed within the 24 h after death. All the fish were subjected to linear biometry (total length, total weight) and dissected to recover right and left fillets that were freeze dried for further analytical purposes. The feed distributed in the last rearing period was also collected for each sampling site (except for the extensive farms), and the main hydrological characteristics were also monitored during the experimental period.

### 2.2. Composition Analysis

Dry matter (AOAC method 934.01), protein by the Kjeldahl method (AOAC 920.53), and lipid [60] contents were determined in the fresh and minced left fillets without skin obtained after dissection and on the minced feeds. To analyze the lipid fraction and de-lipidize the sample (residual from chloroform: methanol extraction) for their stable isotope ratio (SIRA) we followed the previously mentioned procedure recovering the residue after lipid extraction of the freeze-dried sample.

### 2.3. Isotopic Measurements, Standards and Equations

The stable isotope ratios (^13^C/^12^C, ^15^N/^14^N, ^18^O/^16^O and ^2^H/^1^H) were measured on dried samples obtained after pooling 5 fish fillet muscle samples per farm. The values for δ^13^C and δ^15^N were measured by continuous flow (ConfloII) elemental analysis isotope ratio mass spectrometry (CF-EA-IRMS) using an EA 1108 CHN elemental analyzer (ThermoFisher, Milan, Italy) (oxidation column temperature: 1050 °C; reduction column temperature: 650 °C; and gas chromatography column: 65 °C) coupled to a DeltaPlus mass spectrometer (ThermoFisher, Milan, Italy). The values for δ^18^O and δ^2^H were measured by continuous flow (ConfloII) total combustion elemental analysis isotope ratio mass spectrometry (CF-TC-IRMS) using a TC/EA (ThermoFisher, Milan, Italy) (pyrolysis column temperature: 1450 °C and gas chromatography column: 45 °C) coupled to the previously described IRMS instrument. As fish muscle has a C/N ratio of less than 5:1, the CF-EA-IRMS system operated in the dual isotope mode, allowing δ^13^C and δ^15^N to be measured on the same sample. Complete feeds were analyzed in single mode, meaning that δ^13^C and δ^15^N were analyzed in separate analyses due to the different ratio of C/N for each diet and, therefore, different quantities were weighed for each one. The results of carbon (δ^13^C) and nitrogen (δ^15^N) isotope ratio analyses were reported in per mill (‰) on the relative δ-scale and referred to the following international standards: V-PDB (Vienna Pee Dee Belemnite) for the carbon isotope ratio and atmospheric air for the nitrogen isotope ratio. In the same way, the oxygen (δ^18^O) and hydrogen (δ^2^H) isotope ratio analyses were reported in per mill (‰) on the relative δ-scale and referred to the international standard V-SMOW (Vienna Standard Mean Ocean Water) for both oxygen and hydrogen isotope ratios.

All the results were calculated according to the following equation,
Delta (‰) = [(R_Sample_/R_Reference_) − 1] × 1000(1)
where R is the ratio of the heavy to light stable isotope (e.g., ^15^N/^14^N) in the sample (R_Sample_) and in the standard (R_Reference_).

The precision (standard deviation) of the analysis of laboratory standard (urea) for δ^13^C was ± 0.11‰ (*n* = 10) and ± 0.15‰ for δ^15^N (*n* = 10). To evaluate the precision of the fish muscle sample analyses, one sample was repeatedly measured (*n* = 10) with a calculated standard deviation of ± 0.09‰ for δ^13^C and ±0.12‰ for δ^15^N. The remaining muscle samples were analyzed in triplicate with a standard deviation lower than 0.15‰ for δ^13^C and 0.20‰ for δ^15^N measurements. In addition, the diets and their ingredients were analyzed in triplicate with a standard deviation lower than 0.21‰ for δ^13^C and 0.27‰ for δ^15^N measurements. International standard USGS-40 (δ^13^C = 26.39‰ and δ^15^N = 4.52‰) was analyzed at the beginning and at the end of each run to check the instrument functioning. Moreover, one sample of fish muscle was calibrated against the international reference materials IAEA-CH7 (δ^13^C = 32.15‰) and IAEA-CH6 (δ^13^C = 10.4‰) for carbon and IAEA-N1 (δ^15^N = 0.45‰) and IAEA-N2 (δ^15^N = 20.39‰) for nitrogen ratios, respectively, and used as the working standard. This working standard was analyzed at regular intervals in each run to control the repeatability and to correct the possible drift deviations in the measurements. In the same way, international standard IAEA-601 Benzoic acid (δ^18^O = 23.3‰) and IAEA-CH7 Polyethylene (δ^2^H = 100.3‰) were analyzed at the beginning and end of each run to check the instrument functioning when measuring δ^18^Oand δ^2^H. The standard deviation of the measurements (*n* = 10) determined using the respective reference gas was ±0.10‰ for δ^18^O and ± 0.8‰ for δ^2^H. Each sample of fish muscle was analyzed in triplicate to obtain data representative of the material. The standard deviation for the analyses was <0.5‰ for δ^18^O and <3.0‰ for δ^2^H.

### 2.4. Statistics

A one-way analysis of variance (ANOVA) was applied to study the differences among the farming systems. The level of significance was set at *p* < 0.05. The Duncan post hoc test was used for comparison of means. All statistical analyses were performed using SPSS/PC Release 17 for Windows (SPSS Inc., Chicago, IL, USA).

## 3. Results and Discussion

### 3.1. Biometric Measurements and Proximate Composition

The size distribution of the fish sampled is presented in Table 2 and reveals an unbiased size distribution among the different farming systems in the 11 fish farms. The average fish weight varied from 552 g to 663 g and the average total length of the fish varied from 36.3 cm to 38.0 cm. The farming system significantly affected the chemical composition of the fillet (Table 2). The fillets of sea bass from the extensive systems exhibited a lower lipid content (2.7%) and higher moisture levels, making them readily distinguishable from the intensively reared fish. The fish reared in cages presented a significantly lower fat content in fillets than the fish kept in intensive land-based farms (7.1% vs. 8.9%).

### 3.2. Isotope Ratio Values in Feed and Animal Tissues

#### 3.2.1. Influence of the Defatted Process on Isotopic Data of Fish Muscle Tissues

As lipids are depleted in ^13^C relative to the other major components (protein and carbohydrate), the variation in lipid content affects the δ^13^C of the whole muscle and in some cases the δ^15^N value. For this reason, defatted muscle is generally considered to enable more accurate and precise tissue comparison [61]. The effect of lipid extraction on δ^15^N remains unclear due to the limited number of studies and inconclusive results obtained in marine organisms [62,63,64,65]. According to Serrano et al. [47], no significant differences were observed in δ^15^N values between tissues with and without lipids. Additionally, the same authors demonstrated that the influence of lipid depletion in ^13^C is dependent on the tissues studied. δ^13^C values are clearly correlated with the lipid content of the tissue, liver, red muscle, and gills presenting significant differences in δ^13^C value when lipids are removed, while no significant differences were observed for the white muscle and gonads. In order to clarify the importance of the ^13^C depletion of lipids in our dataset, nine samples from the C4 farm (fish in sea cages) were extracted and two subsamples were obtained from each one: lipids and defatted muscle. An analysis was performed on the δ^13^C, δ^15^N, and δ^2^H values for the whole muscle sample, the lipid fraction extracted from the muscle, and the defatted muscle, the results being shown in Table 3. In our case no significant differences were observed in δ^15^N values between tissues with and without lipids. The depletion in ^13^C linked to lipids can be observed in the lower δ^13^C values obtained for lipid fraction and the significant differences found for δ^13^C values between tissues with and without lipids. Similar conclusions can be drawn for the δ^2^H values. The lipid fraction shows an important depletion in ^2^H and therefore significant differences can be observed in the δ^2^H values between the whole and the defatted muscle samples. In this research, the well documented δ^13^C contribution was witnessed and the contribution of δ^2^H has been studied for the first time; the discrimination of the diet and geographical origin of whole muscle from European sea bass was still possible and no lipid extraction was performed in order to avoid this time-consuming step.

#### 3.2.2. Discrimination of Farming System

The muscle from extensively farmed fish (E) exhibited less negative δ^13^C values compared to fish reared intensively inland (I) and in sea cages (C) (Table 2); this makes it possible to discriminate the farming system and is consistent with previous data found in literature based on wild and farmed fish discrimination [14,46,66,67]. The differences in the isotopic values of the three fish rearing groups were clearly reflected in significantly different carbon and nitrogen isotope values in the sea bass fillets (Table 2). The average values for δ^13^C were −16.17‰ for extensively reared sea bass, and −21.87‰ and −21.52‰ for sea bass intensively reared in tanks inland and in sea cages, respectively. The average δ^15^N values obtained were 14.02‰ for the extensively reared sea bass, and 10.38‰ and 11.40‰ for intensive sea bass reared inland in tanks and in cages in the sea, respectively.

According to the studies published by DeNiro et al. [27,28], the slight enrichment in δ^13^C and δ^15^N of the animal’s tissues compared with the values found in their diets was named trophic shift. In extensive farming, fish growth is limited by the availability of the natural food supply, and fasting has been reported to affect the isotopic composition of different species. In fact, the amount of food consumed, as well as its dietary composition, are known to affect the trophic shifts of C and N in fish [61,68,69,70]. The δ^13^C values of the sea bass from the E3 farm, an extensive system with diet supplementation, placed them inside the “Intensive population” (Figure 1). The use of a commercial diet had measurable consequences on the isotopic composition of fish muscle that was more important than other growing parameters such as physical activity (extensive system). Meanwhile, a significant increment in δ^13^C and δ^15^N values was found in the muscle of the rest of extensive compared to the intensive farmed fish (Figure 1), in agreement with previous findings in wild turbot [71]. While the isotopic values of the sea bass reared in intensive farming systems remained mostly constant, the δ^13^C values for the extensive farming fish displayed a significant variation (Figure 1). This may be explained by the controlled food type and dose applied to the intensive farming, as long as the wild sea bass faces with different food availability with scarcity periods into which the fish need to use its body stores. Therefore, farmed fish is expected to show more negative values of δ^13^C than the wild fish, since its diet is less variable and richer in fat, resulting in muscles with higher lipid content that induces larger isotopic fractionation of δ^13^C [71]. The δ^15^N values depend on the content, origin, and type of proteins supplemented during the diet of fish. The main natural diet of wild sea bass is hard crabs and various small fishes, depending upon the geographical location. However, protein from different origins (animal or vegetable) can be supplemented during the diet of farmed sea bass, so that the variation in δ^15^N values found between wild and intensive sea bass may be influenced by the natural or administered diet [37,71].

In our experimental work, the metabolic assimilation of the feed was reflected in the muscle of the fish with the characteristic trophic shift, defined as the difference between the mean muscle isotopic signature of a group and that of its respective diet. The Δδ^13^C and Δδ^15^N values (Table 4 and Figure 2) found for fish reared intensively inland ranged from 1.07‰ to 2.24‰ and from 2.95‰ to 4.98‰, respectively. The values found for fish reared intensively in sea cages ranged from 1.10 to 1.96 for δ^13^C and from 3.73 to 4.05 for δ^15^N. Finally, the values for the trophic shift found for the “semi-extensive” group (E3) of fish were 0.72‰ and 5.63‰ for carbon and nitrogen, respectively. The δ^13^C and δ^15^N enrichment between muscle and diet values obtained in the present study was in agreement with the literature [27,28,72,73,74], 1–3‰ for δ^13^C and 1.3–5.3‰ for δ^15^N. Of note were the sea bass reared “semi-extensively” as the values found for their trophic shift were slightly different to the ranges found in literature. The explanation for those differences is probably due to the rearing conditions of those animals, whose artificial diet was also supplemented by natural prey found in the extensive location in which they were reared [75]. Clear differences in the trophic shift were also observed within both intensive farming systems (sea cages and inland tank) (Figure 2). The temperature regime of the growth media and food deprivation due to differences on feeding frequency and quantity of the intensive systems may affect the fish isotopic values [76].

#### 3.2.3. Geographical Origin Discrimination

Regarding the geographical origin of the extensively farmed fish samples, the δ^2^H and δ^18^O ratios of the sea bass muscle presented powerful markers. The wild samples belonging to the southern location (37.50° N) showed the most enriched δ^2^H and δ^18^O values (Figure 3) compared to the northern locations (44.57° N, 44.96° N, and 45.76° N). Both isotope ratios for fish muscle are dominated by the isotopic composition of the water ingested by the fish [39,67] and controlled by regional climatic conditions and temperature [77]. The differences in δ^2^H and δ^18^O for the muscle samples are therefore reflecting the different geographical latitudes of the three locations. Therefore, understanding the spatial distribution of these stable isotopes to determine the geospatial origin of water gives valuable information to provide evidence for geographical differentiation.

Additionally, geographical discrimination for intensively reared sea bass was attempted based on the isotopic signatures. A geographical correlation was found for the fish reared under the intensive system in sea cages between their origin and their hydrogen isotope values (Figure 3). These results demonstrated the strong correlation between the δ^2^H ratio of the muscle, and the hydrogen isotope composition of the sea water as previously concluded for the extensively reared sea bass. However, no patterns were identified for the fish reared under intensive inland tank systems and their isotope signatures, probably due to the complex water supply, which is often a mixture of marine, inland, and well sources. In Appendix A, two maps of isotope ratios in annual average precipitation for hydrogen and oxygen are shown. In general terms, δ^2^H and δ^18^O values were less negative from the southern to the northern of Italy and the coastal areas. As it can be seen in Figure 3, this behavior was reflected in the isotopic values of fish reared under the same system (sea cages or extensively one). However, some differences were observed between both systems, which displayed a constant shift of about 30–35 δ^2^H values in sea bass from near geographical areas in spite of showing similar δ^2^H of water obtained from the precipitation map, with the exception of the “false declared” extensive E3 (Appendix A and Figure 3), whose hydrogen isotopic value was more similar to the fish reared in the intensive sea cages. This behavior may be explained by the nature of this farming system, since the sea bass from E3 reared in lagoon water are expected to be more affected by evapotranspiration and precipitation in comparison to the sea bass reared in the sea. Meanwhile, the cage systems account for about 6–8 m deep, limiting the habitat of sea bass which can be found for up to 15 m deep in extensive systems, exposing them to a higher contribution of evapotranspiration and precipitation too. In a similar way, a depletion of δ^18^O was found in the sea bass muscle from the intensive systems in sea cages, which could also be related with the fractionation induced by a higher influence of the environmental factors. In this case, the sea bass sampled from E3 farming displayed the same behavior than the rest of extensive systems, with the exception of E4, which were more similar to the intensive systems in sea cages. Whilst water can be considered as the only hydrogen source of the isotopic ratio of hydrogen, oxygen also depends on the carbon dioxide and atmospheric oxygen [78] that could explain this different behavior.

The identification of differences in the isotope values of samples from different origins therefore shows the potential of stable isotope analysis for the verification of geographic traceability. However, the identified patterns should be confirmed in further studies with a significantly larger number of samples from certified origins and over a longer period. The small geographical latitude range covered by our experiment (36.70° N to 45.80° N) supports the potential of a combination of isotopes to trace the origin of products on a larger scale and gains more importance when taking into account the current regulations regarding providing consumers with additional information.

## 4. Conclusions

The results of the current study showed the viability of δ^13^C and δ^15^N to discriminate cultured sea bass from different farming systems (extensive vs. intensive) reared at different geographical sites in Italy. The metabolic assimilation of commercial diets led to an enrichment of the δ^13^C and δ^15^N of the fish muscle, reflecting a characteristic trophic shift. Additional information was obtained from δ^18^O and δ^2^H, which made it possible to differentiate extensively and intensively (in cages) farmed sea bass. Finally, a depletion of δ^13^C and δ^2^H linked to the lipid fraction of the samples was observed, while no significant differences were found for δ^15^N values between tissues with and without lipids.

## Figures and Tables

**Figure 1 animals-10-02042-f001:**
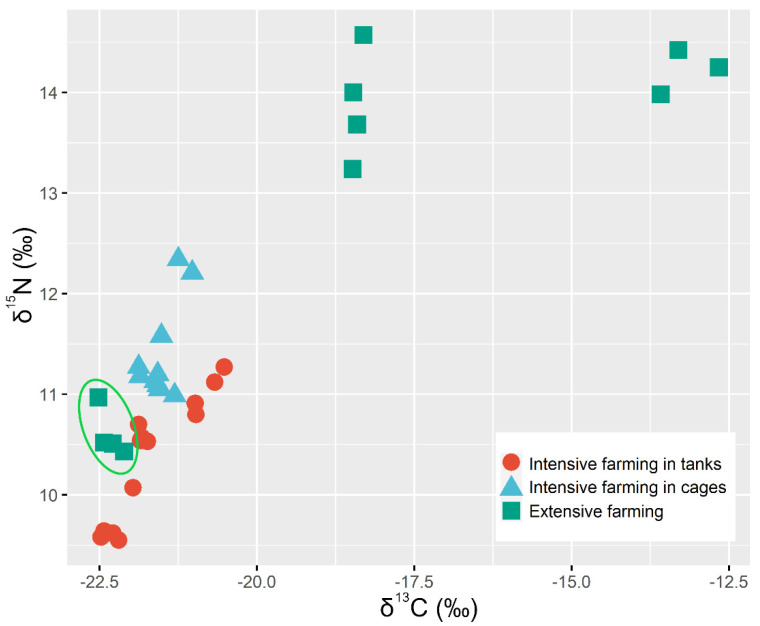
δ^13^C and δ^15^N values in sea bass muscle. Each data point represents the measure of a pooled sample of 5 fish fillets.

**Figure 2 animals-10-02042-f002:**
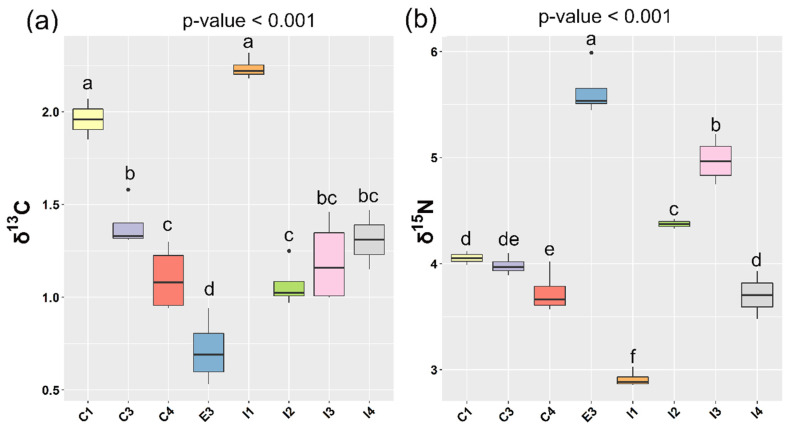
Trophic shift measured for carbon (**a**) and nitrogen (**b**) isotopic ratios between the feed supplied and muscle of sea bass. A one-way ANOVA was performed on the data and the letters above the boxes correspond to the Tukey Honestly Significant Difference (HSD) post hoc test.

**Figure 3 animals-10-02042-f003:**
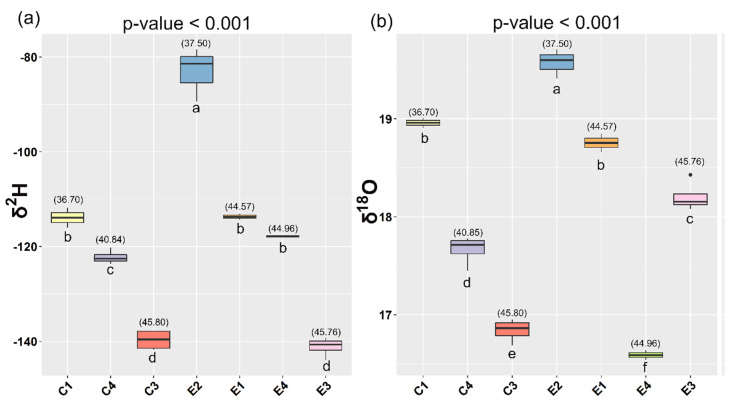
δ^2^H (**a**) and δ^18^O (**b**) values (‰) of muscle of E. sea bass reared in extensive systems and sea cages. The latitude of each group is shown in parenthesis. A one-way ANOVA was performed on the data and the letters below the boxes correspond to the Tukey HSD post hoc test.

**Table 1 animals-10-02042-t001:** General farms characteristics, water quality, commercial feed producer, and number of fish sampled.

Farm Code	Latitude	Longitude	Farming System	Water Source	Temperature (°C)	Salinity (‰)	Feed Producer	Sampled Fish
C1	36.70° N	15.12° E	*Sea cage*	Sea	18	37	Biomar	10
C3	45.80° N	13.55° E	*Sea cage*	Sea/river	12	34	Skretting	20
C4	40.84° N	17.47° E	*Sea cage*	Sea	9.8	37	Aller	20
E1	44.57° N	12.33° E	*Extensive*	Sea	16	28	---	10
E2	37.50° N	12.48° E	*Extensive*	Sea	16	37	---	15
E3	45.76° N	13.17° E	*Semi-intensive*	Lagoon	8	20	Skretting	20
E4	44.96° N	12.32° E	*Extensive*	Sea	16	25	---	10
I1	42.42° N	11.28° E	*Inland in pond*	Well	18	24	Skretting	20
I2	40.93° N	14.03° E	*Inland outdoor*	Well	18	28	Biomar	20
I3	44.95° N	12.32° E	*Inland outdoor*	Sea	17	28	Skretting	20
I4	37.98° N	12.51° E	*Inland outdoor*	Sea	18	37	Biomar	10

**Table 2 animals-10-02042-t002:** Effect of the rearing system (E = Extensive; I = Intensive inland; C = Intensive in sea cages) on biometric traits, chemical composition, and δ^13^C and δ^15^N of the muscle of the sea bass.

	E	I	C	MSE
*Biometric traits*				
n. of samples	55	70	50	
Whole body weight (g)	633.2 ± 136.4	613.7 ± 175.4	552.5 ± 146.2	30,964.2
Total length (cm)	38.0 ± 4.22	36.5 ± 3.40	36.3 ± 2.21	10.776
*Chemical composition* (g/100 g)				
n. of samples	55	70	50	
Moisture	76.25 ± 1.79 ^a^	69.54 ± 2.57 ^c^	71.64 ± 3.17 ^b^	7.004
Protein Content	19.21 ± 0.41 ^b^	19.73 ± 1.13 ^a^	19.31 ± 0.73 ^b^	0.875
Lipid	2.67 ± 1.80 ^c^	8.91 ± 2.79 ^a^	7.13 ± 3.01 ^b^	7.361
*Stable isotope data* (‰)				
n. of samples	11	14	10	
δ^13^C	−16.17 ± 2.81 ^a^	−21.87 ± 0.64 ^b^	−21.52 ± 0.26 ^b^	1.927
δ^15^N	14.02 ± 0.45 ^a^	10.38 ± 0.57 ^b^	11.40 ± 0.49 ^b^	0.312

Stable isotope data were measured on pooled samples of 5 fish fillets. MSE = Mean Square Error. Mean values in the same row with different letters differ significantly (*p* < 0.01).

**Table 3 animals-10-02042-t003:** Mean isotopic values of whole muscle, lipid fraction, and defatted muscle of nine sea bass from C4 farm.

Measure	Whole Muscle	Lipid Fraction	Defatted Muscle
**δ**^13^C (‰)	−21.74 ± 0.19	−26.05 ± 0.17	−19.88 ± 0.11
**δ**^15^N (‰)	11.12 ± 0.24		11.01 ± 0.22
**δ**^2^H (‰)	−124.43 ± 5.52	−197.78 ± 2.37	−85.52 ± 1.05

**Table 4 animals-10-02042-t004:** δ^13^C and δ^15^N trophic shifts for intensive (inland and sea cages) or semi-intensive (E3) reared sea bass.

Farming System	Farm Code	δ^13^C (‰) Diet	δ^13^C (‰) Group (Mean Value)	Δδ^13^C (‰)	δ^15^N (‰) Diet	δ^15^N (‰) Group (Mean Value)	Δδ^15^N (‰)
***I***	I1	−24.06	−21.82	2.24	7.67	10.62	2.95
	I2	−23.45	−22.38	1.07	5.22	9.60	4.38
	I3	−21.98	−20.79	1.19	6.05	11.03	4.98
	I4	−23.44	−22.13	1.31	6.14	9.84	3.70
***C***	C1	−23.10	−21.14	1.96	8.22	12.27	4.05
	C3	−22.89	−21.50	1.39	7.10	11.08	3.98
	C4	−22.82	−21.72	1.10	7.56	11.29	3.73
***E***	E3	−23.05	−22.34	0.72	4.98	10.61	5.63

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
