# Peer review of "The Use of Stable Isotope Ratio Analysis to Trace European Sea Bass (D. labrax) Originating from Different Farming Systems"

_animals, 2020, doi:10.3390/ani10112042_

Round 1
Reviewer 1 Report
Thank you for picking up our suggestions, the results can be much better understood now.
Author Response
We clearly appreciate the reviewer's comment because it clearly enhanced the quality of this paper.
Reviewer 2 Report
Dear Authors,
The modifications made to the manuscript have significantly increased the readability of the research and as such made it more attractive to the reader. The discussions added are very good and the manuscript now flows much better. However, I have noticed that some minor formatting changes are required before the manuscript may be accepted for publication. Please see for details below.
Line 29: sentence is incomplete and was not modified as stated. It should probably read "..made it possible to distinguish the geographical origin..." rather than "...made it possible the geographical discrimination..."
Line 168: there should be a space between 650 and the degree sign
Figure 2. the y-axes should read δ13C and δ15N rather than "delta13C" and "delta 15N"
Line 332: a full stop is missing at the end of the sentence "...evidence for geographical differentiation."
Line 353: it should be either "In a similar way" or "in the same way" but not "in a similar same way"
Figure 3. the y-axes should read δ2H and δ18O rather than "delta H" and "delta O" that do not even give the isotope but only the element
Author Response
Reviewer 2:
Dear Authors,
The modifications made to the manuscript have significantly increased the readability of the research and as such made it more attractive to the reader. The discussions added are very good and the manuscript now flows much better. However, I have noticed that some minor formatting changes are required before the manuscript may be accepted for publication. Please see for details below.
Line 29: sentence is incomplete and was not modified as stated. It should probably read "..made it possible to distinguish the geographical origin..." rather than "...made it possible the geographical discrimination..."
This sentence has been modified as pointed out in line 29.
Line 168: there should be a space between 650 and the degree sign
A space between 650 and ºC has been included in line 168.
Figure 2. the y-axes should read δ13C and δ15N rather than "delta13C" and "delta 15N"
A new Figure 2 with the y-axes renamed as reviewer suggests has been included in line 320 of the manuscript.
Line 332: a full stop is missing at the end of the sentence "...evidence for geographical differentiation."
A full stop has been included in line 331 as reviewer suggests.
Line 353: it should be either "In a similar way" or "in the same way" but not "in a similar same way"
The sentence was corrected in line 352.
Figure 3. the y-axes should read δ2H and δ18O rather than "delta H" and "delta O" that do not even give the isotope but only the element
A new Figure 2 with the y-axes renamed as reviewer suggests has been included in line 367 of the manuscript.

This manuscript is a resubmission of an earlier submission. The following is a list of the peer review reports and author responses from that submission.
Round 1
Reviewer 1 Report
The paper presents a nice case study combining N & C isotopes to trace the feeding system and O and H isotopes for geographical origin determination. However, the sample size per farm is really low (35 fish in total from 11 farms, yielding about three per farm), so I have the feeling there needs to be a bit more careful interpretation of results. However, the tendencies are obviously visible, although it is hrd to test them statistically.
I do not understand the sentence: "Five freeze dried fillet muscle sub samples were pooled to obtain the number of samples per provider to be analyzed for the stable isotope ratios:...how many individuals? to get pseudoreplicates of 3 homogenized samples? This must be much more clearly described.
Line 84: Dicentrarchus labrax L. in Italics
Line 212: in Table 2 the number of fish per treatment for lenght measurement is lacking - also the abbreviations should be explained in the table captions.
How were the data created for each farm? Now only one value with no uncertainties is represented.
The description of results clearly lack the link to the number of fish that have been used for a pooled sample, etc. - and no uncertainties are provided.
In Fig. 1 - what are the data points? Different fish per farm? Different pooled samples per fish?
Please provide the correlation between feed values of N and C - as this seems not clear only from table 3, which also has no caption. Also only 8 farms show up in this table, and 11 were mentioned in the beginning.
Compare the H isotope values in fish fillet with environmental data to provide the evidence for geographical differntiation, there are global maps of O and H isotopes existing.
Fig. 2 and Fig 3 need a more detailed caption and also n of fish considered. This is nowhere reported.
Overall, the paper is interesting, but lacks significant clarity in the description of number of fish per site, how pooled samples were produced. Also later specific relations, e.g. between feed N and C values and fillet are not well presented. This needs to be discussed and shown in more detail. Overall, the number of fish seems very low, although the tendencies are clearly visible. I would also like to see a better discussion on natural values of O and H exisiting in the water in thes region, there are global maps existing. However, this might also follow a seasonal pattern. This is also not discussed. This means the representation of the method, the data & their relations and discussion need to be clearly improved. However, with a really improved and more clear representation of the sample sizes, the critical link between Feed and Fillet values of N and C, and an improved discussion of geographically typical values of O and H with fillets, and their potential seasonal changes, the paper is a first step in combining these indicators. It has to be noted that the nubmer of fish is far too low for being of real value for a serious model.
Please point out the limitations (in numbers of fish and also the maybe weak relation between C and N isotopes in feed with fillet values - why did you not present a figure/statistics showing this relation?)
Line 325: Additional information was obtained from δ18O and δ2H, which made it possible to differentiate extensively and intensively (in cages) farmed
sea bass. -> I thought
Reviewer 2 Report
Dear authors,
Your article on the use of IRA to distinguish between sea bass production systems and location is interesting and well written. The usefulness of the different elements is well explained and findings are in line with previous research. however, I have same concerns about the description of the materials and methods which is why I recommended major revisions. The materials and methods are not clear enough and need improvement to fully understand and appreciate the research of this paper.
Specific comments are below and I hope they will help the authors with the revision of this manuscript which describes relevant research.
- It might be useful to always have carbon before nitrogen since this is the order in which they are in the period table of elements and also the order in which they are typically presented in papers. This would make it easier for the reader.
- line 29: sentence is incomplete. it should probably read "..made it possible to distinguish the geographical discrimination..."
- line 121 uses incorrect format of reference
- lines 141-142 describe 3 extensive, 3 intensive cage and 5 intensive inland farms. However, later in table 1 this is 4-3-4 (E-C-I); please clarify as it is very confusing
- line 143 references supplementary material. this was not available for download and may have been forgotten to be uploaded?
- Table 1 - please add how many fish were sampled from each site as at present this is not described anywhere in the paper
- lines 154-155 imply that dry matter was determined on freeze-dried samples. I am certain this was not the case. Please rewrite section to clarify for which tests freezer-dried samples and for which fresh/frozen-thawed samples were used.
- lines 161-162 describe how sampling was carried out by pooling sub-samples per fish to account for variations within the fillets. However, later the authors write "to obtain the number of samples per provider to be analysed". this number is never given: there is no information how many fish were collected per provider or how many samples were analysed per provider. Please clarify.
- Section 2.3: all degree signs appear to be underlined. please correct
- line 205 should read "..Dundan post-hoc test..." (t is missing)
- Table 2 - what does MSE stand for? please add to table legend or text
- Table 2 - please add number of samples for which the average result is presented
- Section 3.2.1 does not discuss how the δ13C and δ15N of extensively reared fish compare to literature. this would be very interesting and useful information (absolute values not trophic shift).
- Figure 2 - why are the results of E3 not presented in this graphic? Also, it might be useful to combine figure 2 and 3 add notes into the figure for latitude. this would make it better comparable with extensive farming
- section 3.2.3 is extremely confusing. the purpose is clear: defatting is a lot of work and its impact on all parameters (N, H, O) is not always well known. As fat content varies amongst individual fish and because fat is very different as compared to protein, defatted samples are measured for C and N. The purpose now was to investigate the impact of H and O. However, the conclusion is not well presented (did defatting have an impact on H and O???). Also, lines 315-316 say that no lipid extraction was performed. was this for all elements or only H (O appears not to have been investigated??)? Since I assume this study was performed prior to analysing all samples, it should be first in the results rather than last as I assumed that all samples were defatted up to this point. Please make this a lot clearer as it is an important point. how can this paper be cited in the future if the author is not sure whether or not SIR presented are for defatted fish or whole fish?
